# The Impact of Gleason Grade 3 as a Predictive Factor for Biochemical Recurrence after Robot-Assisted Radical Prostatectomy: A Retrospective Multicenter Cohort Study in Japan (The MSUG94 Group)

**DOI:** 10.3390/medicina58080990

**Published:** 2022-07-25

**Authors:** Makoto Kawase, Shin Ebara, Tomoyuki Tatenuma, Takeshi Sasaki, Yoshinori Ikehata, Akinori Nakayama, Masahiro Toide, Tatsuaki Yoneda, Kazushige Sakaguchi, Jun Teishima, Kazuhide Makiyama, Takahiro Inoue, Hiroshi Kitamura, Kazutaka Saito, Fumitaka Koga, Shinji Urakami, Takuya Koie

**Affiliations:** 1Department of Urology, Gifu University Graduate School of Medicine, Gifu 5011194, Japan; buki2121@gifu-u.ac.jp; 2Department of Urology, Hiroshima City Hiroshima Citizens Hospital, Hiroshima 7308518, Japan; shinbone0127@yahoo.co.jp; 3Department of Urology, Yokohama City University, Yokohama 2360004, Japan; tatenuma@yokohama-cu.ac.jp (T.T.); makiya@yokohama-cu.ac.jp (K.M.); 4Department of Nephro-Urologic Surgery and Andrology, Mie University Graduate School of Medicine, Tsu 5148507, Japan; t-sasaki@clin.medic.mie-u.ac.jp (T.S.); tinoue28@clin.medic.mie-u.ac.jp (T.I.); 5Department of Urology, University of Toyama, Toyama 9300194, Japan; ikehata.y0226@gmail.com (Y.I.); hkitamur@med.u-toyama.ac.jp (H.K.); 6Department of Urology, Dokkyo Medical University Saitama Medical Center, Koshigaya 3438555, Japan; akinori@dokkyomed.ac.jp (A.N.); kzsaito@dokkyomed.ac.jp (K.S.); 7Department of Urology, Tokyo Metropolitan Cancer and Infectious Diseases Center Komagome Hospital, Tokyo 1138677, Japan; m.toide@gmail.com (M.T.); f-koga@cick.jp (F.K.); 8Department of Urology, Seirei Hamamatsu General Hospital, Hamamatsu 4308558, Japan; yonet@sis.seirei.or.jp; 9Department of Urology, Toranomon Hospital, Tokyo 1058470, Japan; kazhimaro@gmail.com (K.S.); shinji.urakami@toranomon.gr.jp (S.U.); 10Department of Urology, Kobe City Hospital Organization Kobe City Medical Center West Hospital, Kobe 6530013, Japan; teishimaj@yahoo.co.jp

**Keywords:** multicenter cohort study, biochemical recurrence, Gleason grade, intermediate-risk prostate cancer, robot-assisted radical prostatectomy

## Abstract

*Background and Objectives*: This study’s objective was to examine patients treated with robot-assisted radical prostatectomy (RARP) for intermediate-risk prostate cancer (IR-PCa), and to identify preoperative risk factors for biochemical recurrence (BCR) in these patients in Japan. *Materials and Methods*: We conducted a retrospective multicenter cohort study of patients with PCa who underwent RARP at 10 institutions in Japan. A total of 3195 patients were enrolled in this study. We focused on patients with IR-PCa who underwent RARP. We obtained data on pre- and postoperative covariates from the enrolled patients. Biochemical recurrence-free survival was the primary endpoint of this study. We also identified useful preoperative predictive factors for BCR in patients with IR-PCa after RARP. *Results*: A total of 1144 patients with IR-PCa were enrolled in this study. The median follow-up period was 23.7 months. At the end of the follow-up period, 94 (8.2%) patients developed BCR. The 2 and 3 year biochemical recurrence-free survival (BRFS) rates were 92.2% and 90.2%, respectively. Using the Kaplan–Meier method, Gleason grade (GG) 3 was significantly associated with poor BRFS compared with ≤GG 2. In multivariate analysis, GG 3 was a significant predictive factor for BCR in patients with IR-PCa. *Conclusions*: The results of the study indicated a significant relationship between GG 3 and post-RARP BCR in patients with IR-PCa.

## 1. Introduction

According to several guidelines, radical prostatectomy (RP) is recognized as the definitive treatment modality for clinically localized or locally advanced prostate cancer (PCa) [1,2]. Robot-assisted RP (RARP), a minimally invasive procedure, may be advantageous in reducing bleeding and transfusion rates [3]. Indeed, RARP has been widely accepted as a surgical procedure for the management of non-metastatic PCa [4], with its use greatly increasing over the past decade. Risk classification has been proposed as a method for evaluating the probability of recurrence and prognosis of PCa [5]. Regarding D’Amico risk stratification, intermediate-risk is defined as follows: clinical stage ≤T2b, a biopsy Gleason score (GS) of 7, or a prostate-specific antigen (PSA) level of >10 and ≤20 ng/mL [6]. In general, patients with intermediate-risk PCa (IR-PCa) receive various definitive therapies, including active surveillance, RP, external beam radiotherapy (EBRT), or brachytherapy, because the majority of patients experience IR-PCa as a localized disease [7]. Nonetheless, several studies reporting biochemical recurrence (BCR) rates after definitive treatment for IR-PCa had varying results, with 5 year BCR rates ranging between 2% and 70% [8]. Although several authors have identified that preoperative PSA, PSA density (PSAD), prostate volume, clinical and pathological T stage, pathological GS, and positive surgical margin (PSM) were significant predictive factors for BCR in patients with PCa who underwent RP, many of them were analyzed using postoperative covariates [9,10,11]. To date, the Gleason grading system remains a strong prognostic predictor for PCa [12]. Rather than using the Gleason grading system, ranging from 2 to 10, which has been accepted by the World Health Organization for the 2016 edition of *Pathology and Genetics and Other Guidelines*, the new grading system and the terminology Grade Group 1–5 were proposed by the 2014 International Society of Urological Pathology (ISUP) consensus conference [13]. Although analyses of both biopsy and surgical specimens have shown that GS 3 + 4 and 4 + 3 tumors differ in prognosis [12,14], IR-PCa is complex and heterogeneous, and there is a significant difference in the 5 year BCR rate between favorable and unfavorable IR-PCa [15]. Thus, this study used preoperative covariates, especially the biopsy Gleason grade (GG), to identify predictive factors for BCR in patients treated with RARP for IR-PCa.

## 2. Materials and Methods

### 2.1. Patient Population

This study was conducted with the approval of the Institutional Review Board of Gi-fu University (authorization number: 2021-A50). The requirement for patient consent was waived because of the retrospective study design. According to the provisions of the ethics committee and ethics guidelines in Japan, the study information is disclosed to the public in the case of retrospective and/or observational studies, with materials, such as existing documentation. The retrospective multicenter cohort study for robot-assisted radical prostatectomy. Available online (4 August 2021): https://www.med.gifu-u.ac.jp/visitors/disclosure/docs/2021-B039.pdf (accessed on 25 January 2022).

A retrospective multicenter cohort study was conducted in patients who underwent RARP for PCa between September 2012 and August 2021 at 10 institutions in Japan. Preoperative information included patient age, height, weight, Eastern Cooperative Oncology Group performance status (ECOG-PS) [16], preoperative serum PSA level, prostate volume, biopsy GG, clinical T stage, and D’Amico risk stratification. In all patients, tumor staging was based on the American Joint Committee on Cancer’s 8th edition cancer staging manual [17]. In this study, no data were collected according to whether the enrolled patients underwent the evaluation using magnetic resonance imaging before the prostate biopsy. Biopsy GG was evaluated according to the International Society of Urologic Pathology’s (ISUP) 2014 guidelines [13]. Briefly, the GG is classified into five grades according to the ISUP grading system as follows: GG 1 (GS ≤ 6), with only individual discrete well-formed glands; GG 2 (GS 3 + 4 = 7), with predominantly well-formed glands with a lesser component of poorly formed/fused/cribriform glands; GG 3 (GS 4 + 3 = 7), with predominantly poorly formed/fused/cribriform glands with a lesser component of well-formed glands; GG 4 (GS 8), with either only poorly formed/fused/cribriform glands, predominantly well-formed glands with a lesser component of lacking glands, or predominantly lacking glands with a lesser component of well-formed glands; GG 5 (GS 9 or 10), with no gland formation (or with necrosis), with or without poorly formed/fused/cribriform glands.

All patients enrolled in this study were treated with RARP. Decisions regarding pelvic lymph node dissection (PLND), extent of PLND, and the nerve-sparing approach were left to the discretion of individual surgeons or each institution.

### 2.2. Pathological Analysis

All prostatectomy specimens were sectioned according to the whole-mount staining technique and evaluated according to the ISUP 2014 guidelines [13]. Pathologists shaved the prostate apex perpendicular to the prostatic urethra. Subsequently, the pathologists coned the bladder neck margin from the specimen and sectioned it perpendicularly. Regarding the remaining prostate tissue, complete sectioning was performed at 3 mm intervals, along a plane perpendicular to the urethral axis.

### 2.3. Follow-Up Schedule

In all patients, serum PSA and testosterone levels were assessed at 3 month intervals after surgery. Patients were categorized as having disease recurrence or PSA failure when the postoperative serum level of PSA rose to 0.2 ng/mL. If postoperative PSA levels were not below 0.2 ng/mL, the date of RP was defined as the time of disease recurrence.

### 2.4. Endpoints and Statistical

In this study, BRFS was the primary endpoint. The secondary endpoint was the determination of the association between BCR and preoperative covariates. The software JMP 14 (SAS Institute Inc., Cary, NC, USA) was used for data analysis. The Kaplan–Meier method was used to evaluate BRFS after RARP. The relationship between BCR and subgroup classification was analyzed using the log-rank test. A Cox proportional hazards model was used for multivariate analysis. All *p*-values were two-sided, and *p*-values < 0.05 were considered statistically significant.

## 3. Results

### 3.1. Patient Characteristics

Table 1 summarizes patient demographics. This study enrolled 3195 patients. Among them, 1908 patients (59.7%), including 367 (11.5%) with low-risk PCa, 1609 (50.4%) with high-risk PCa, and 3 (0.09%) with missing data with IR-PCa were excluded from the study. Finally, the oncological outcomes and perioperative complications of 1144 patients (35.8%) were analyzed. Although multiple surgeons performed RARP at each institution, the surgeon volume was not evaluated in this study.

### 3.2. Oncological Outcomes

Patients were followed up for a median period of 23.7 months (interquartile range, 10.8–45.2 months). At the end of the follow-up period, 94 patients (8.2%) were diagnosed with BCR and six (0.5%) had developed radiographic recurrence. There were no cases of castration-resistant PCa or deaths from PCa among the enrolled patients. Thirteen patients (1.1%) died from unrelated causes (details unknown). The BRFS rate was 92.2% at 2 years and 90.2% at 3 years.

Figure 1 shows the relationship between BCR and D’Amico risk stratification, according to preoperative PSA levels, biopsy GS, and clinical T stage. The 2 year BRFS rate was 92.3% for clinical T1b/c/T2a and 91.3% for clinical T2b (*p* = 0.596; Figure 1A). Furthermore, it was 92.8% for initial PSA levels of <10 ng/mL, 90.6% for ≥10 ng/mL (*p* = 0.012; Figure 1B), 94.5% for biopsy GG ≤2, and 87.6% for biopsy GG ≥3 (*p* = 0.001; Figure 1C).

In multivariate analysis, GG3 was identified as a significant predictive factor for BCR in patients with IR-PCa who underwent RARP (Table 2).

## 4. Discussion

According to the D’Amico risk stratification, the majority of patients with PCa are diagnosed as having intermediate-risk PCa rather than low- or high-risk PCa [10,12,18]. A potential problem is that patients with IR-PCA have heterogeneous biological and clinical features with distinct therapeutic outcomes [10]. According to Keane et al., PCa-specific mortality varies significantly according to the subgroup of patients with IR-PCa who underwent EBRT [19]. Similarly, contemporary patients with IR-PCa treated with RP can be classified into favorable and unfavorable subgroups, with a significant difference in postoperative biochemical outcomes [12]. Based on these results, novel predictive factors may be needed to enable more precise tumor characterization, therapeutic decision-making, and individualized therapeutic approaches.

Several studies in patients with PCa who underwent RARP have reported various predictive factors for BCR [9,10,11,20,21]. A single center study involving 784 Japanese patients treated with RARP for localized PCa has reported that there were significant associations of BCR with PSAD, pathological T stage, GS, and PSM [9]. Based on 944 patients with PCa who underwent RARP at Karolinska University Hospital with a median follow-up period of 6.3 years, preoperative PSA > 10 ng/mL, pathological GS ≥ 4 + 3, pathological T3 disease, PSM, and lower surgeon volume were associated with an increased risk of BCR in multivariate analysis [20]. According to 1384 patients with PCa who had BCR after RARP with a median follow-up period of 5 years, the strongest predictors of BCR were a pathological GS ≥ 8 and a pathological stage of T3b/T4 [21]. Therefore, it may be difficult to predict BCR after surgery using preoperative covariates, particularly IR-PCa.

Analysis of the predictive factors using preoperative variables revealed that the PSAD was a useful predictive factor for adverse pathology and/or BCR and had a superior ability to predict the aggressiveness and prognosis of localized PCa in patients treated with RP [22]. Narita et al. found that BRFS was most strongly predicted by a PSAD of <0.6 ng/mL/cm^3^ in patients with IR-PCa [10]. Additionally, patients with IR-PCa who had a PSAD of ≥0.6 ng/mL/m^3^ were at a risk of BCR equivalent to that of patients with high-risk PCa [10]. Multivariate analysis for patients with IR-PCa revealed that a valuable predictor for post-RP BRFS was a preoperative PSAD of <0.3 ng/mL/m^3^, but that PSA was not [23]. According to PSAD, the cut-off value was identified as 0.5 using the area under the receiver operating characteristic curve (AUC). However, AUC was 0.572 with a sensitivity of 24.2% and a specificity of 89.4%. Therefore, the median PSAD was used as the cut-off value in this study. Although PSA-related factors, such as PSAD, were not significantly different in this study, these factors may play an important role in predicting BCR after definitive therapies in patients with PCa.

Recent reports showed that GS 4 + 3 reflects a poorer prognosis for patients with IR-PCa, when compared with GS 3 + 4 [12,18,23,24]. Zumsteg et al. divided patients with IR-PCa who underwent EBRT into favorable or unfavorable risk groups and compared BRFS between them [18]. Favorable risk was defined as a GS of ≤3 + 4, <50% of biopsy cores containing cancer, and a single National Comprehensive Cancer Network criterion [18]. Patients in the unfavorable risk group had poor BRFS, risk of distant metastasis, and PCa-specific mortality [18]. Regarding patients with IR-PCa who underwent RP based on the aforementioned classification system, those with unfavorable risk had significantly higher rates of advanced pathological stage and BCR than those in the favorable-risk group [12]. Several studies have investigated cribriform architecture in IR-PCa; however, the impact of cribriform architecture in GS ≥ 4 + 3 is less well established [25,26]. In addition, IR-PCa is a heterogeneous disease, and this heterogeneity may lead to significant interobserver variability in tumor grading [27]. Therefore, it may be difficult to evaluate the association between biopsy GG and BCR in patients with IR-PCa using a multicenter, retrospective study.

The present study has certain limitations. One limitation was the retrospective nature of the multicenter data, which may have conferred a susceptibility to potential selection bias, as diagnostic and surgical approaches varied between the included institutions. Additionally, the follow-up period of the study was fairly short, which may have hampered the ability to investigate the relationship between variables due to the high risk attributed to PSA. Third, we did not determine the positive biopsy core rate. Fourth, GG was not re-evaluated in all biopsy specimens by a single pathologist in this study. Finally, adjuvant treatment was administered depending on the primary physician’s and/or the patient’s preference.

## 5. Conclusions

The results from the present study demonstrated GG 3 to be a valuable predictive factor for post-RARP BCR in patients with IR-PCa. Indeed, GG may be applied as a simple, helpful preoperative factor in a real-world clinical setting to differentiate heterogeneous patients with IR-PCa, thereby allowing different prognostic groups to be treated according to their individualized treatment planning and counseling. The findings of this study should be validated using a larger prospective dataset.

## Figures and Tables

**Figure 1 medicina-58-00990-f001:**
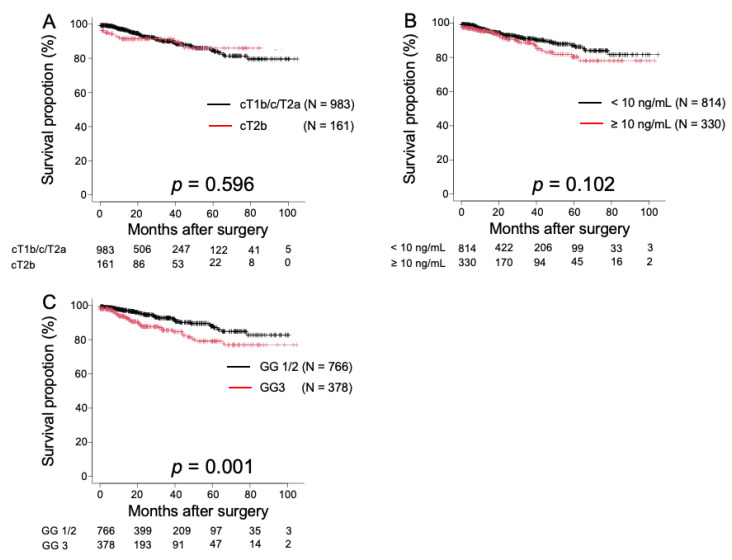
Kaplan–Meier estimates of biochemical recurrence-free survival (BRFS) according to clinical T stage (**A**), initial prostate-specific antigen (PSA) levels, which were stratified by a cutoff of 10 ng/dL (**B**), and the biopsy Gleason grade (GG) (**C**). The 2 year BRFS rates were 92.3% and 91.3% in patients with clinical T1/2a and clinical T2b, respectively (*p* = 0.596; Figure 1A); 92.8% and 90.6% in patients with an initial PSA level of <10 ng/mL and ≥10 ng/mL, respectively (*p* = 0.102; Figure 1B); and 94.5% and 87.6% in patients with biopsy GG of 1/2 and GG of 3, respectively (*p* = 0.001; Figure 1C).

**Table 1 medicina-58-00990-t001:** Patient demographics.

Variables	
Age (year, median, IQR)	68 (64–72)
BMI (median, IQR)	23.6 (21.7–25.6)
ECOG Performance Status (number, %)	
0	1111 (97.1)
1	31 (2.7)
2	2 (0.2)
Initial PSA (ng/mL, median, IQR)	7.3 (5.4–10.4)
Prostate volume (mL, median, IQR)	30.0 (22.4–40.0)
PSAD (ng/mL/cm^3^, median, IQR)	0.25 (0.17–0.36)
Biopsy Gleason score (number, %)	
3 + 3	121 (10.6)
3 + 4	645 (56.4)
4 + 3	378 (33.0)
Clinical T stage (number, %)	
1b	3 (0.3)
1c	282 (24.7)
2a	698 (61.0)
2b	161 (14.1)
Pathological T stage (number, %)	
2	881 (77.0)
3a	203 (17.7)
3b	55 (4.8)
4	2 (0.2)
x	3 (0.3)
Pathological N stage (number, %)	
0	771 (67.4)
1	16 (1.4)
Not evaluated	357 (31.2)
Surgical margins status (number, %)	
Negative	800 (69.9)
Positive	315 (27.5)
Follow-up period (months, median, IQR)	23.7 (10.8–45.2)

Abbreviations are as follows: QR, interquartile range; BMI, body mass index; ECOG, Eastern Cooperative Oncology Group; PSA, prostate-specific antigen; PSAD: prostate-specific antigen density.

**Table 2 medicina-58-00990-t002:** Multivariate analysis.

Variables	*p* Value	Odds Ratio	95% Confidence Interval
Gleason grade			
≥3	0.002	1.906	1.255–2.894
Prostate volume (mL)			
≥30	0.066	1.568	0.972–2.530
PSAD (ng/mL/cm^3^)			
≥0.25	0.089	1.574	0.932–2.658
Initial PSA (mg/mL)			
≥10	0.456	1.205	0.738–1.969
ECOG-PS			
≥1	0.156	0.238	0.033–1.732
Age (year)			
≥68	0.751	1.072	0.699–1.643
Body mass index			
≥23.5	0.785	1.060	0.696–1.615
Clinical T stage			
≥2b	0.807	1.072	0.612–1.879

Abbreviations are as follows: PSAD, prostate-specific antigen density; PSA, prostate-specific antigen; ECOG-PS, Eastern Cooperative Oncology Group performance status.

## Data Availability

The data presented in this study are available on request from the corresponding author. The data are not publicly available due to privacy and ethical reasons.

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
