# Peer review of "The Impact of Gleason Grade 3 as a Predictive Factor for Biochemical Recurrence after Robot-Assisted Radical Prostatectomy: A Retrospective Multicenter Cohort Study in Japan (The MSUG94 Group)"

_medicina, 2022, doi:10.3390/medicina58080990_

Round 1

Reviewer 1 Report

Title:

The title reflects the aims of the study well.

Abstract:

The abstract reflects their work and the aims.

Introduction:

Introduction is well-written and easy to follow.

Materials and Methods:

・Did authors re-evaluate all prostate biopsy specimens and prostatectomy specimens, some of which predate the introduction of Gleason grade?

・In 2.2., authors demonstrated that ‘Surgeons shaved the prostate apex perpendicular to the prostatic urethra. Subsequently, the surgeons coned the bladder neck margin from the specimen and sectioned it perpendicularly. Regarding the remaining prostate tissue, complete sectioning was performed at 3-mm intervals, along a plane perpendicular to the urethral axis.’ Is it true?

Results:

The results were clearly presented and could be followed easily in tables.

There are some points that I would suggest.

・In Table1, authors should add the data of pathological T stage, N stage and PSM rate.

・How many patients had the MRI scan? If many, then PIRADS is required in this study that emphasize preoperative examinations.

・Did authors analyze surgeon volume?

・Why did authors use the PSAD cut-off of 0.25? In Discussion section, authors introduce the cut-off of 0.3 or 0.6.

Conclusions:

The conclusion reflects their work in a plausible way.

Author Response

14, July, 2022

Dear Editor-in-Chief, the Medicina

Thank you very much for the opportunity to resubmit our manuscript. The comments of the reviewers have been helpful in allowing us to revise our manuscript according to the reviewers’ recommendations. It would be our great pleasure if you would take our manuscript into consideration for publication in the Medicina.

I look forward to your reply.

Best regards,

Takuya Koie, M.D.

Professor

Department of Urology, Gifu University Graduate School of Medicine

1-1 Yanagido, Gifu, Japan

Tel: +81-582306000

FAX: +81582306339

E-mail: goodwin@gifu-u.ac.jp

Response to Reviewer 1

The authors appreciate the reviewer’s comments. The authors’ point-by-point responses to the comments are given below.

Title:

The title reflects the aims of the study well.

Abstract:

The abstract reflects their work and the aims.

Response:

Thank you for your kind comments.

Materials and Methods:

・Did authors re-evaluate all prostate biopsy specimens and prostatectomy specimens, some of which predate the introduction of Gleason grade?

Response:

The authors have added the following sentence on line 216.

Fourth, GG was not re-evaluated in all biopsy specimens by a single pathologist in this study.

・In 2.2., authors demonstrated thatSurgeons shaved the prostate apex perpendicular to the prostatic urethra. Subsequently, the surgeons coned the bladder neck margin from the specimen and sectioned it perpendicularly. Regarding the remaining prostate tissue, complete sectioning was performed at 3-mm intervals, along a plane perpendicular to the urethral axis.’ Is it true?

Response:

The authors have revised the following sentences on line 109:

Surgeons Pathologists shaved the prostate apex perpendicular to the prostatic urethra. Subsequently, the surgeons pathologists coned the bladder neck margin from the specimen and sectioned it perpendicularly.

Results:

The results were clearly presented and could be followed easily in tables.

There are some points that I would suggest.

・In Table1, authors should add the data of pathological T stage, N stage and PSM rate.

Response:

The authors have added these data to Table 1 according to the reviewer’s recommendation.

・How many patients had the MRI scan? If many, then PIRADS is required in this study that emphasize preoperative examinations.

Response:

The authors have added the following sentence on line 92:

In this study, no data were collected according to whether the enrolled patients underwent the evaluation using magnetic resonance imaging before the prostate biopsy.

・Did authors analyze surgeon volume?

Response:

The authors have revised the following sentence on line 134:

Although multiple surgeons performed RARP at each institution, the surgeon volume was not evaluated in this study.

・Why did authors use the PSAD cut-off of 0.25? In Discussion section, authors introduce the cut-off of 0.3 or 0.6.

Response:

The authors have revised the following sentence on line 189:

According to PSAD, the cut-off value was identified as 0.5 using the area under the receiver operating characteristic curve (AUC). However, AUC was 0.572 with a sensitivity of 24.2% and a specificity of 89.4%. Therefore, the median PSAD was used as the cut-off value in this study.

Conclusions:

The conclusion reflects their work in a plausible way.

Response:

Thank you for your kind comments.

Reviewer 2 Report

The study used the gleason grade 3 as a predictive factor for biochemical recurrence after robot-assisted radical prostatectomy. The patients number (3,195) enrolled in this study and the median follow-up period (23.7 months) meet the statistical criteria. The author used the Kaplan–Meier method to find that Gleason grade (GG) 3 was significantly associated with poor BRFS, which has important clinical implications for prognosis after robot-assisted radical prostatectomy.  the authors consider making the following revisions

1)       A full name list of abbreviations should be provided at the end of manuscript because there are so many abbreviations in the manuscript such as BRFS, RARP, BCR, PSA et al., though the full names of this abbreviations have been provided somewhere in the manuscript. But it is more convenient if such an abbreviation list can be provided.

2)       The lines of Figure 1 were confusing. Maybe the author can consider use different colors and shapes to represent clinical T stage, prostate-specific antigen, Gleason grades.

3)       A histological picture to illustrate what is gleason grade should be provided

4)       Table 1 645 (56.4 should be corrected

Author Response

14, July, 2022

Dear Editor-in-Chief, the Medicina

Thank you very much for the opportunity to resubmit our manuscript. The comments of the reviewers have been helpful in allowing us to revise our manuscript according to the reviewers’ recommendations. It would be our great pleasure if you would take our manuscript into consideration for publication in the Medicina.

I look forward to your reply.

Best regards,

Takuya Koie, M.D.

Professor

Department of Urology, Gifu University Graduate School of Medicine

1-1 Yanagido, Gifu, Japan

Tel: +81-582306000

FAX: +81582306339

E-mail: goodwin@gifu-u.ac.jp

Response to Reviewer 2

The authors appreciate the reviewer’s comments. The authors’ point-by-point responses to the comments are given below.

The study used the Gleason grade 3 as a predictive factor for biochemical recurrence after robot-assisted radical prostatectomy. The patients’ number (3,195) enrolled in this study and the median follow-up period (23.7 months) meet the statistical criteria. The author used the Kaplan–Meier method to find that Gleason grade (GG) 3 was significantly associated with poor BRFS, which has important clinical implications for prognosis after robot-assisted radical prostatectomy.  the authors consider making the following revisions.

1) A full name list of abbreviations should be provided at the end of manuscript because there are so many abbreviations in the manuscript such as BRFS, RARP, BCR, PSA et al., though the full names of this abbreviations have been provided somewhere in the manuscript. But it is more convenient if such an abbreviation list can be provided.

Response:

The authors have added the abbreviations on line 233.

Abbreviations: RP, radical prostatectomy ; PCa, prostate cancer ; RARP, robot-assisted radical prostatectomy; GS, Gleason score; PSA, prostate-specific antigen ; IR-PCa, intermediate-risk prostate cancer; EBRT, external beam radiotherapy; BCR, biochemical recurrence; PSAD, prostate-specific antigen density; PSM, positive surgical margin; ISUP, International Society of Urological Pathology; GG, Gleason grade; ECOG-PS, Eastern Cooperative Oncology Group performance status; PLND, pelvic lymph node dissection; BRFS, biochemical recurrence-free survival.

2) The lines of Figure 1 were confusing. Maybe the author can consider use different colors and shapes to represent clinical T stage, prostate-specific antigen, Gleason grades.

Response:

The authors have revised Figure 1 according to the reviewer’s recommendation.

3) A histological picture to illustrate what is Gleason grade should be provided.

Response:

The authors could not include histological pictures due to copyright issues. We have added the following sentence on line 94.

Biopsy GG was evaluated according to the International Society of Urologic Pathology (ISUP) 2014 guideline [13]. Briefly, the GG is classified into five grades according to the ISUP grading system as follows: GG 1 (GS ≤6), only individual discrete well-formed glands; GG 2 (GS 3 + 4 = 7), predominantly well-formed glands with a lesser component of poorly formed/fused/cribriform glands; GG 3 (GS 4 + 3 = 7), predominantly poorly formed/fused/cribriform glands with a lesser component of well-formed glands; GG 4 (GS 8), either only poorly formed/fused/cribriform glands, predominantly well-formed glands with a lesser component of lacking glands, or predominantly lacking glands with a lesser component of well-formed glands; GG 5 (GS 9 or 10), no gland formation (or has necrosis) with or without poorly formed/fused/cribriform glands.

4) Table 1 645 (56.4 should be corrected.

Response:

The authors have corrected Table 1.

Round 2

Reviewer 1 Report

I think we have corrected the problem sufficiently.

This manuscript is a resubmission of an earlier submission. The following is a list of the peer review reports and author responses from that submission.